# A Grid Search-Based Multilayer Dynamic Ensemble System to Identify DNA N4—Methylcytosine Using Deep Learning Approach

**DOI:** 10.3390/genes14030582

**Published:** 2023-02-25

**Authors:** Rajib Kumar Halder, Mohammed Nasir Uddin, Md. Ashraf Uddin, Sunil Aryal, Md. Aminul Islam, Fahima Hossain, Nusrat Jahan, Ansam Khraisat, Ammar Alazab

**Affiliations:** 1Department of Computer Science and Engineering, Jagannath University, Dhaka 1100, Bangladesh; 2School of Information Technology, Deakin University, Geelong 3125, Australia; 3Department of Computer Science and Engineering, Hamdard University Bangladesh, Munshiganj 1510, Bangladesh; 4Department of Computer Science and Engineering, Eastern University, Dhaka 1345, Bangladesh; 5School of IT, Melbourne Institute of Technology, Melbourne 3000, Australia

**Keywords:** DNA N4-Methylcytosine, deep learning, word embedding, grid search, natural language processing

## Abstract

DNA (Deoxyribonucleic Acid) N4-methylcytosine (4mC), a kind of epigenetic modification of DNA, is important for modifying gene functions, such as protein interactions, conformation, and stability in DNA, as well as for the control of gene expression throughout cell development and genomic imprinting. This simply plays a crucial role in the restriction–modification system. To further understand the function and regulation mechanism of 4mC, it is essential to precisely locate the 4mC site and detect its chromosomal distribution. This research aims to design an efficient and high-throughput discriminative intelligent computational system using the natural language processing method “word2vec” and a multi-configured 1D convolution neural network (1D CNN) to predict 4mC sites. In this article, we propose a grid search-based multi-layer dynamic ensemble system (GS-MLDS) that can enhance existing knowledge of each level. Each layer uses a grid search-based weight searching approach to find the optimal accuracy while minimizing computation time and additional layers. We have used eight publicly available benchmark datasets collected from different sources to test the proposed model’s efficiency. Accuracy results in test operations were obtained as follows: 0.978, 0.954, 0.944, 0.961, 0.950, 0.973, 0.948, 0.952, 0.961, and 0.980. The proposed model has also been compared to 16 distinct models, indicating that it can accurately predict 4mC.

## 1. Introduction

DNA methylation is an epigenetic modification in which chromatin structure, DNA orientation, DNA integrity, inactivation of the X chromosome, regulation of gene expression, cell differentiation, cancer development, and DNA–protein interactions are altered, keeping the original gene sequence unmodified. This modification plays a significant role in developmental and pathological processes, such as aging, carcinogenesis, genomic imprinting, transposable elements repression, X chromosome inactivation, etc. [1,2,3]. On the other hand, changes in DNA methylation cause several illnesses including tumorigenesis, abnormalities of imprinting, cardiovascular diseases, autoimmune diseases, neurological diseases, cancer, etc. [1,4]. N4-Methylcytosine (4mC) occurs at the C4 position of cytosine in both prokaryotic and eukaryotic cells. It is responsible for participation in the restriction–modification system to provide a bacterial immune response against occupied DNA, as well as DNA repair, expression, or replication. As for experimental methods and data, current knowledge about other biological functions of 4mC is insufficient, and the dataset used to identify 4mC and studies related to 4mC is limited [5]. Although there are several approaches to identifying 4mC methylation, such as mass spectrometry, methylation-precise PCR, single-molecule real-time (SMRT) sequencing, and 4mC-Tet-assisted bisulfite, those experiments on the functional or biological behavior of 4mC sites are expensive, time-consuming, and fail to identify 4mC areas with time efficiency when applied to large sequencing data [5,6,7,8]. Recently, machine learning approaches have been applied to identify 4mC sites with promising outcomes. Machine learning-based approaches have enhanced 4mC identification research and have been highly successful in predicting 4mC sites. The success of machine learning-based techniques (i.e., their predictive power) in distinguishing 4mC sites from non-4mC sites is highly dependent on the quality of the extracted features [1]. Deep learning is also used extensively in studying proteins, DNA sequences, and RNA sequences as a powerful and popular tool in machine learning [9]. Ensemble methods in machine learning have also been used in several studies to detect 4mC sites. Although the state-of-the-art methods have continuously yielded promising findings, their lack of generalizability necessitates the creation and development of new prediction algorithms to accurately detect 4mC sites. Therefore, a reliable prediction model for detecting 4mC sites at a large scale in a gene is highly desirable to fully understand the biological role of 4mC, as 4mC may play a supporting role in genetic stabilization, regeneration, and development. The following summarizes the aims and major contributions of this research work to achieving the objectives:To develop a grid search-based weighted average ensemble (WAE) system for identifying DNA N4-Methylcytosine where learned information is transmitted along from a layer to the next (details are provided in Section 4).The model applies multi-configuration 1D CNN to perform the ensemble method in each layer to improve prediction efficiency and find the best results, in addition to transferring learning knowledge from one layer to another.A grid search-based weighted average ensemble (WAE) technique has been applied to find the optimal accuracy at each layer.We trained the model with eight large publicly accessible datasets and compared its performance with the existing models.

The remaining paper is organized as follows. Section 2 describes tasks related to the determination of DNA N4-methylcytosine, existing methods, and available techniques. In Section 3, we give details about our proposed methodology. Section 4 contains the experimental results, as well as a discussion of the proposed model. Section 5 ends with a conclusion and future work.

## 2. Literature Review

To identify 4mC sites, several experimental studies have been conducted. We evaluated existing works in this section by analyzing these related works, and the limitations of these works are identified and documented in depth below.

S. Zhang et al. (2022) [10] proposed a multi-source feature and gradient boosting decision tree-based identification model of DNA N4-methylcytosine sites. They proceeded by extracting features from original sequences using multi-source feature representation methods, which include the mononucleotide binary and K-mer frequency, dinucleotide binary and position-specific frequency, ring-function hydrogen-chemical properties, dinucleotide-based DNA properties, and trinucleotide-based DNA properties. Following this, a gradient boosting decision tree was used to find the optimal feature set and to remove redundant information. Finally, a support vector machine was utilized to predict whether a site was 4mC or not.

Yu. Lezheng et al. (2022) [11] proposed a convolutional recurrent neural network model to identify DNA N4-methylcytosine. For representing DNA sequences, they considered one-hot and dictionary encoding methods. Following that, three representative deep learning algorithms with different network architectures for classification were chosen, including a convolutional neural network (CNN), a recurrent neural network (RNN), and a recurrent neural network (RNN) with bidirectional long-short-term memory cells (BiLSTM).

L. Wang et al. (2022) [12] proposed 4mCPred-FSVM, a tool that uses a fuzzy model to identify DNA N4-methylcytosine sites. They used position-specific trinucleotide propensity to construct the feature vectors, which were then fed into the FSVM (fuzzy support vector machine) to build the final model.

J. Jin et al. (2022) [13] proposed a deep learning model for predicting DNA N4-methylcytosine sites in the mouse genome. This network architecture is composed of three major segments: (A) the adaptive embedding module, (B) the GRU module, and (C) the classification module. The adaptive embedding module was created to tokenize an original DNA sequence and obtain appropriate embeddings adaptively using the encoding matrix. Following that, the GRU module was then utilized to extract long and short distant information in sequence. The classification module used a max-pooling layer to determine which feature in each GRU unit is the most relevant or important.

J. Khanal et al. (2021) [3] proposed a deep learning model to identify DNA N4-methylcytosine sites in the Rosaceae genome, relying on distributed feature representation. In the first stage, the K-mer (k = 3) feature encoding technique was used to represent each DNA sequence as a fixed length of words. Then, the ‘word2vec’ embedding method was applied to map each word to its corresponding vector form. The authors used convolutional neural networks (CNN) in the second stage to classify 4mCs and non-4mCs based on selected features.

A. Wahab et al. (2021) [5] proposed an identification model of N4-methylcytosine using deep learning, and natural language processing. The K-mer (k = 2, 3, 4) feature encoding technique was used to transfer the combination of nucleotides (A, C, G, T) into the sequence of words in this model. The ‘word2vec’ embedding method was applied to present each word in the vector form. A continuous bag-of-word (CBOW) approach was employed to train the word2vec model. Finally, a CNN was used to classify the 4mC and non-4mC sites. Three different k-fold cross-validations (three-fold, five-fold, and ten-fold) were applied to carry out the preeminent identification.

G. Fang et al. (2021) [6] proposed a word2vec based deep learning network to predict DNA N4-methylcytosine sites. Using the word2vec embedding method, each feature derived from K-mers (K = K = 3, 4, 5, 6) was presented in vector form. The feature matrix sequence was then fed into 3-CNN to categorize the 4mCs and non-4mCs. 

M. Tahir et al. (2021) [9] proposed an intelligent and robust computational prediction model for DNA N4-methylcytosine sites via natural language processing. The authors examined two input sequence representations. The initial encoding method was one-hot encoding, which encoded each DNA sequence with (1, 0, 0, 0), (0, 1, 0, 0), (0, 0, 1, 0), and (0, 0, 0, 1) for A, C, G, and T, respectively. The distributed feature representation learnt by the word2vec model, on the other hand, was the second encoding approach. Then, new word representation was fed into the convolution neural network (CNN) model. The CNN model has various layers, including a convolution layer, pooling layer, ReLU layer, normalization layer, dropout layer, fully connected layer, etc.

H. Zulfiqar et al. (2021) [14] proposed Deep-4mCW2V, where a sequence-based predictor was used to identify N4-methylcytosine sites. In this model, the training data were first converted into numerical feature vectors using the “word2vec” technique. Following this, the feature vectors were fed into the 1-D CNN for classification. A 10-fold CV was applied to split the entire dataset into 10 groups of relatively equal size.

D.Y. Lim et al. (2021) [15] developed the iRG-4mC tool, which is based on the CNN-LSTM model. The DNA sequence was encoded using a combination of one-hot encoding and nucleotide chemical properties in the proposed system (NCP). The final sequence was transmitted into the LSTM for feature optimization, and three fully connected layers were used for the final outcome.

M.M. Hasan et al. (2020) [8] proposed a model named i4mC-Mouse to identify DNA N4-methylcytosine sites in the mouse genome using multiple encoding schemes. The authors used six encoding techniques, namely K-space nucleotide composition (KSNC), K-mer nucleotide composition (K-mer), mononucleotide binary encoding (MBE), dinucleotide binary encoding (DBE), electron–ion interaction pseudopotential (EIIP), and dinucleotide physicochemical composition (DPC), to represent a DNA sequence as fixed-length feature vectors. The WR feature selection method was used to remove the noisy feature. Finally, five machine learning classifiers were employed to differentiate between 4mC and non-4mC sites, namely random forest (RF), Bayesian network (NB), support vector machine (SVM), K-nearest neighbor (KNN), and AdaBoost (AB). A 10-fold cross-validation was applied to partition the whole data into training and testing sets.

A. Wahab et al. (2020) [16] proposed a model named DNC4mC-Deep to identify DNA N4-methylcytosine sites based on different encoding schemes by using deep learning. Deep learning techniques were used to classify a DNA sequence that was represented as a fixed length of feature vectors. To generate methylcytosine samples, six types of feature encoding methods were used: binary encoding (BE), DNC (2-mer), TNC (3-mer), multivariate mutual information (MMI), nucleotide chemical property (NCP), and nucleotide chemical property and nucleotide frequency (NCPNF).

Z. Zhao et al. (2020) [17] proposed a model of DNA N4-methylcytosine sites via boost-learning various types of sequence features. The DNA sequence was first encoded with one-hot binary (OHB), sequential nucleotide frequency (SNF), K-nucleotide frequency (KNF), K-spectrum nucleotide pair frequency (KSNPF), and PseDNC. Top-ranked features from the XGBoost training procedure were chosen. Finally, the chosen features were used to train the support vector machine (SVM) classification model. A K-fold cross-validation (K = 10) was used to split the dataset into training and test sets.

J. Khanal et al. (2019) [18] proposed an identification model of N4-methylcytosine sites in prokaryotes using a convolutional neural network. In this model, every DNA sequence was represented as a binary vector using a one-hot encoding technique. The K-fold cross-validation technique was used to split the datasets into the training and testing sets and, finally, CNN was employed to classify the 4mCs and non-4mCs sequences.

B. Manavalan et al. (2019) [19] proposed a sequence-based meta-predictor for accurate DNA 4mC site prediction using effective feature representation. They began by generating 56 probabilistic features using four ML algorithms: SVM, random forest [RF], gradient boosting [GB], and extremely randomized tree [ERT], as well as seven feature encodings: K-mer composition, binary profile [BPF], dinucleotide binary profile encoding [DPE], local position-specific dinucleotide frequency [LPDF], ring-function-hydrogen-chemical properties [RFHC], and dinucleotide physicochemical composition. They subsequently fed these probabilistic features into a SVM to construct a final prediction model.

Limitations: All the aforementioned models are unable to expand their current knowledge from their resources. We have proposed a multilayer dynamic system in which the learning knowledge flows from layer to layer, and every layer can achieve an optimal accuracy that enhances the overall model’s performance. The related studies covered in this section is summarized in Table 1 with respect to diverse attributes.

## 3. Materials and Methods

### 3.1. Dataset Construction

The benchmark and independent datasets in this paper are collected from different sources [3,5,6,8,9,16,17,18,19] to measure the efficiency of the proposed model for a fair comparison with current predictors. These datasets contain eight different species, namely *Caenorhabditis elegans (C. elegans)*, *Drosophila melanogaster (D. melanogaster)*, *Arabidopsis thaliana (A. thaliana)*, *Escherichia coli (E. coli)*, *Geoalkalibacter subterraneus (G. subterraneus)*, *Geobacter pickeringii (G. pickeringi)*, *Fragaria vesca (F. vesca)*, and *Rosa chinensis (R. chinensis)*. All samples are 41 bp long, with the 4mC site in the middle. The threshold of CD-HIT was adjusted at 80% in order to eliminate redundant sequences and to prevent the predictor from overfitting. As a result, the number of negative samples will outnumber the number of positive samples. An identical number of negative samples were picked at random from the eight species in order to equalize the positive and negative samples. Therefore, these datasets can be written in the following form:(1)$$S_{i}=S_{i}^{+}\cup S_{i}^{-}\;\mathrm{Where}\;i=1,2,3,4,.......,8$$
where $S_{i}$ represents the total number of positive and negative samples for the eight species. The eight positive datasets from the various species are included in the subsets $S_{i}^{+}$(i=1,2,3,4,.......,8); $S_{i}^{-}$ (i=1,2,3,4,.......,8) contains the negative samples. In set theory, the union is represented by the symbol ∪. Table 2 lists the specifics of the eight benchmark datasets.

### 3.2. Feature Representation

A DNA sequence is represented as a fixed length of feature vectors which can be classified by deep learning algorithms. We used the K-mer approach to break the genomic sequence into fixed-length words (3-mer). It is a standard feature encoding algorithm widely used in various prediction tasks. When K = 2, we call this method dinucleotide composition (DNC), and when K = 3, we call this method trinucleotide composition (TNC). In this study, we have used K = 3. Trinucleotide composition in K-mers is commonly used to predict DNA N4-methylcytosine (4mC) using deep learning approaches, because 4mC modification tends to occur in specific patterns within DNA sequences, and these patterns are often related to the surrounding trinucleotide context of the 4mC site. In K-mer analysis, the K-mer size is the length of the substring that is extracted from the DNA sequence for analysis. Trinucleotide composition specifically refers to K-mers of length 3, where the substring consists of three adjacent nucleotides. By using trinucleotide composition in K-mers of length 3, we can capture information about the surrounding nucleotides in a DNA sequence that are most relevant for predicting 4mC modification. This is because the presence of 4mC is known to be influenced by the nucleotides that immediately precede and follow the modification site. Therefore, analyzing trinucleotide composition allows the model to capture these important contextual features in the data. In addition, using trinucleotide composition in K-mers of length 3 also reduces the total number of K-mers that need to be considered in the analysis, compared to using larger K-mer sizes. This can improve the efficiency of the analysis and reduce the computational requirements for training the deep learning model. In TNC, all samples of 41 nt produce 39 components with the equation of L − k + 1. Here, L stands for the sequence length, and k stands for the K-mer value as an integer [11,12,13,14,15,16,17,18,19,20,21,22,23,24]. ATG, TGC, GCG, and CGA are the four 3-mers that can be tokenized from the DNA sequence “ATGCGA,” for instance.

### 3.3. Distributed Feature Representation

We converted each K-mer word into a 100-dimensional vector format in attempt to acquire discriminating information between the two classes. We used a word embedding approach known as word to vector (word2vec) to transfer the sequences into vector form. This technique generates an optimal set of feature vectors based on the distributional hypothesis. Each word (K-mer) in word2vec is represented by an n-dimensional vector. Here, (L-2) × n array shapes represent the length of each sequence. It is a two-layer neural network that processes text by vectorizing words. It receives input as a text corpus, and its output is feature vectors representing words in that corpus. This technique decreases computational complexity and reduces the noise, ultimately leading to improved performance in the resultant computational model [3,6,25,26]. Either the continuous bag-of-words (CBOW) approach or the skip-gram method can be used to apply the word2vec model. The current word (w(t)) or input is employed in the skip-gram model to predict the surrounding window of context words. The CBOW technique, in contrast, makes an attempt to infer the target word from its nearby (context) words [3]. A five-window CBOW model was developed using the following inputs:(2)$$\sum_{k=2,k\neq 0}^{2}w(t+k)$$

The skip-gram is more practical and produces better results for infrequent words. We used CBOW for word2vec training since in our study we are interested in frequent words. Each word sequence (3-mer) was fed into a word2vec model with two layers. Each sequence of length L was represented by an array of form (L-2) 100, and each word had its own 100-dimension (D) vector representation. As an illustration, the words “ATG” and “TGC” were each represented as a 100-(D) vector of letters $\left[0.14_{1},0.32_{2},0.33_{3},...........,0.11_{100}\right]$ and $\left[0.32_{1},0.14_{2},0.31_{3},...........,0.23_{100}\right]$, respectively. Table 3 provides a list of the word2vec training parameters. The abovementioned three steps are illustrated in Figure 1aa.

### 3.4. Proposed Deep Learning Model

In the final step, the weighted average ensemble technique is adopted on the training and testing set to identify 4mC and minimize generalization error. It combines the predictions from multiple models, where the contribution of each model is weighted proportionally to its capability or skill. As the basic learning model for this work, we used a 1D CNN (deep feed forward neural network) that included several parameters to tune. Afterwards, the convolution neural network (CNN) model was fed the feature vectors as input. Convolution layer, pooling layer, activation layer, normalizing layer, dropout layer, fully connected layer, etc., are some of the layers in the CNN model. The training phase involves adjusting the hyper-parameters. The number of convolution layers, the number of filters within each convolution layer, the size of these filters, and the dropout rate are the hyper-parameters of each CNN. Table 4 displays the ranges of these parameters. Each CNN in the proposed approach is composed of a pair of one-dimensional convolution layers. Each basic learning model has 16 and 32, 32 and 42, and 42 and 64 filters in the first and second layers, respectively. Each model’s first and second layers have filters with sizes of 3 and 5, 5 and 7, and 7 and 9, respectively. A ReLU activation function is applied after each convolution layer. Each convolution layer was followed by a dropout layer with a rate of 0.7 in order to address the overfitting issue. Each model’s two convolution layers’ best features are given to a dropout layer with a probability of 0.5, 0.3, and 0.6 separately, followed by a fully connected layer with one node and a sigmoid function for prediction. Furthermore, the activation layer predicts whether a given DNA sample contains N4-methylcytosine or not, depending on the target class. This layer’s output is normalized to lie within the interval [0,1]. The detailed configurations of the proposed 4mCCNN model are shown in Table 5. This model utilizes a stratified prediction technique. Based on the same training dataset, the three models at each level categorize the same test set. After each layer’s classification is complete, the correctly classified data is added to the pre-training data and compared with the predefined target values in the original data set and inserted into the next layer. In the next iteration, the misclassified data is used as new test data. This process continues until all models have improved performance (TP and TN = 0). However, a complicated aspect of using weighted average ensemble is choosing the relative weight for each ensemble member to achieve the optimal accuracy in each layer. At each layer, a grid search method is used to find the optimal weights for the model, resulting in more “accurate” predictions. The whole approach is called a grid search-based multilayer dynamic system (GS-MLDS). These steps are illustrated in Figure 1bb. The working procedure is demonstrated in Algorithm 1. The overall accuracy for GS-MLDS can be calculated using the following formula:

Number of train data in layer, i: Number of train data in layer (i−1) + Number of TP, TN in layer (i−1).

Test data in layer, i: Number of FP, FN in layer (i−1).

Where i = layer number; i = 1, 2, 3, …, n.
(3)$$TA=\frac{\sum_{i=1}^{n}{\left(TP+TN\right)}_{i}}{S_{Test}(S^{+}+S^{-})}$$
where TA, ${\left(TP+TN\right)}_{i}$, and $S_{Test}$ represent the total accuracy, total number of correctly classified data in each layer, and total number of testing data, respectively. Furthermore, $S^{+}$, and $S^{-}$ represent the total number of 4mC and non 4mC sites, respectively.

In Table 5, the Conv1D (f,s,t) is a one-dimensional convolution operator where f,s, and t stand for the number, size, and stride of filters, respectively. The operator Dropout (p) represents a dropout layer with a probability of p. Dense (n) is a densely connected layer with n nodes. Finally, the sigmoid function is then used to determine if a sequence has a 4mC site or not.
**Algorithm 1:** Proposed Algorithm**Input:****The DNA sequence dataset.****Output:**Classify 4mC and non-4mC 1. Begin  2. Remove the redundant sequences.  3. If total number of (Si+)≠ total number of (Si−) then  5.   Randomly select the equal number of (Si+) and (Si−)
 6. Else  7.   Si=Si+∪Si−   [Where i=1,2,3,4,.......,8]  8. End If  9. Transform each DNA sequence into a fixed length of word.  10. Convert each word into n-dimensional vector form.  11. Split the dataset into training (STrain) and testing (STest) set. Where (STrain) > (STest). The ratio between Si+
   and Si− of each data set is the same as in the entire dataset.  12. Tuning the base classifiers with numerous hyper-parameters.  13. Apply training set to fit each classifier.  14. Integrate the base classifiers and apply gird search to choose the relative weight for each ensemble member.  15. Predict new data.  16. For i = 2 to n do  17.    If TN and TP ≠0 then  18.     STrain = STrain (i−1) + Number of TP, TN in layer (i−1)
 19.     STest = Number of FP, FN in layer (i−1)   [Where i = layer number]  20.    Repeat Step 13 to 15.  21.   Else  22.    Calculate total accuracy using the following formula:        TA=∑i=1n(TP+TN)iSTest(S++S−)   [Where = 1, 2, 3,……., n]  23.   End If  24. End For  25. Stop

## 4. Result and Discussion

The effectiveness of a machine learning model is evaluated using a performance matrix. The “scikit-learn” library’s matrix module contains the methods required to compute performance evaluation metrics. The model’s performance is calculated using a confusion matrix. The confusion matrix produces four results based on the datasets: TP (true positive), TN (true negative), FP (false positive), and FN (false negative). The confusion matrix results of our proposed GS-MLDS are shown graphically in Figure 2a,b. The accuracy, precision, true positive rate, false positive rate, true negative rate, and false negative rate [18] are calculated by employing the following equations:(4)$$\;\mathrm{Accuracy}=1-\frac{R_{-}^{+}+R_{+}^{-}}{R_{}^{+}+R_{}^{-}}$$
(5)$$\;\mathrm{Precision}=\frac{R_{}^{+}}{R_{}^{+}+R_{+}^{-}}$$
(6)$$\;\mathrm{Sensitivity}/\mathrm{True}\;\mathrm{Positive}\;\mathrm{Rate}/\mathrm{Recall}=1-\frac{R_{-}^{+}}{R_{}^{+}}$$
(7)$$\;\mathrm{False}\;\mathrm{Positive}\;\mathrm{Rate}=\frac{R_{+}^{-}}{R_{+}^{-}+R_{}^{-}}$$
(8)$$\;\mathrm{Specificity}/\mathrm{True}\;\mathrm{Negative}\;\mathrm{Rate}=1-\frac{R_{+}^{-}}{R_{}^{-}}$$
(9)$$\;\mathrm{False}\;\mathrm{Negative}\;\mathrm{Rate}=\frac{R_{+}^{-}}{R_{}^{-}}$$
where $R^{+}$ is the total number of 4mC investigated, while $R_{-}^{+}$ is the number of 4mC incorrectly classified as non-4mC sequences. Here, $R^{-}$ is the total number of non-4mC investigated, while $R_{+}^{-}$ is the number of non-4mC incorrectly classified as 4mC.

True positive and true negative are represented as TP and TN in Figure 2a,b above, with the correct number of identified sequences related to 4mC and non-4mC, respectively. False positive and false negative denoted are as FP and FN, respectively, and indicate the false number of identified sequences for 4mC and non-4mC. We have split each dataset 80:20, where 80% is the training set and 20% is the testing set. The total number of 4mC sites correctly identified by this proposed model for *C. elegans*, *D. melanogaster*, *A. thaliana*, *E. coli*, *G. subterraneus*, *G. pickeringi*, *F. vesca*, and *R. chinensis* datasets are 226 out of 231, 337 out of 354, 379 out of 396, 75 out of 78, 177 out of 182, 112 out of 114, 852 out of 865, and 476 out of 485, respectively. The non-4mC sites correctly identified by this proposed model for the same datasets are 222 out of 231, 331 out of 354, 352 out of 396, 74 out of 78, 179 out of 181, 108 out of 114, 799 out of 864, and 461 out of 484, respectively.

ROC curves are a tool that aids in the understanding of probabilistic forecasts for binary (two-class) classifications of predictive modeling problems. The ROC curve demonstrates how well a classification model performs at different thresholds. ROC is a probability curve. TPR and FPR are used to plot the ROC curve, with TPR on the y-axis and FPR on the x-axis. Compared to the existing model, the proposed model outperforms it at classifying and testing datasets when the ROC curve has a lower x-axis and a higher y-axis value. The level or measurement of separability is represented by AUC. The entire two-dimensional region from (0, 0) to (1, 1) below the entire ROC is measured by AUC. A model with a higher AUC predicts 0 s as 0 s and 1 s as 1 s more frequently. The AUC ROC values are graphically presented in Figure 3.

In Figure 3, the blue dots represent the True Positive Rate (TPR) at different classification thresholds, while the orange dots represent the AUC of this proposed model. The proposed model obtained AUC values of 96.97, 94.35, 92.30, 95.51, 98.07, 96.49, 95.49, and 96.70 for the *C. elegans*, D. melanogaster, A. thaliana, E. coli, G. subterraneus, G. pickeringi, F. vesca, and R. chinensis datasets, respectively. That is, the proposed model has 96.97%, 94.35%, 92.30%, 95.51%, 98.07%, 96.49%, 95.49%, and 96.70% probability of distinguishing 4mC from non-4mC for each dataset, respectively. The results of other performance evaluation matrices are shown in Table 6.

The proposed predictor obtained accuracies greater than 90% for each dataset. This model enhances prediction accuracy through layer-by-layer classification techniques. To maximize the accuracy at each layer, we used a grid-based search technique to find the best variety of weights for each layer. For example, Table 7 demonstrate the classification performance of GS-MLDS in each layer. The precision values of this suggested model for each dataset are 96.17%, 93.61%, 89.59%, 94.93%, 98.88%, 94.91%, 92.91%, and 95.39%, respectively. A model’s accuracy specifies how many identified objects are genuinely relented. Precision can also be thought of as the likelihood that a randomly chosen item that is marked as “important” is a true positive, in addition to being a measurement of model performance. Recall and TNR values of more than 90% were obtained for each dataset using the proposed approach. The recall is determined as the proportion of positive samples that were correctly identified as positive to all positive samples. The recall measures how well the model can identify positive samples. The more positive samples that are identified, the larger the recall. This is often referred to as sensitivity or true positive rate. The specificity, also known as the true negative rate, is the likelihood of a negative test if it is actually negative. For each dataset, the corresponding false positive and true negative ratios are 3.89%, 6.49%, 11.11%, 5.12%, 1.10%, 5.26%, 7.52%, and 4.75%, respectively. For each dataset, the false negative and actual positive ratios are 2.16%, 4.80%, 4.29%, 3.8%, 2.7%, 1.75%, 1.5%, and 1.85%, respectively.

Based on experimental results [27], we mentioned, “The appropriately categorized data from one layer may be utilized as a new training set for the following layer. In circumstances, such as these, this training set can reveal new hidden patterns in incorrectly classified data. That is, appropriately identified input from the preceding layer contributes to the model gaining new knowledge”. Although the prior version of the approach gradually improves the accuracy in each layer based on an improved training set, one drawback of this method is that it does not provide guidance on how to achieve maximum accuracy in a layer. As a result, this is a time-consuming process due to the increased number of layers. Furthermore, our main concern in this research work is how to determine the maximum accuracy at each layer to limit the number of additional layers in order to improve the training set to extract new hidden patterns from its resources. We found that each layer’s best combination of weight sets can overcome this limitation. Other researchers commonly used these eight datasets. We have tested the accuracy and compared the proposed model with the existing models on the same independent datasets, as shown in Table 8. We directly submitted the independent datasets to the GS-MLDS. The proposed GS-MLDS yielded accuracies of 0.978, 0.954, 0.944, 0.961, 0.950, 0.973, 0.948, 0.952, 0.961, 0.953, and 0.980. The GS-MLDS outperformed other models with different ratios. GS-MLDS performs well due to the following factors: extending its learning knowledge from its resources to continue the classification process from one layer to another layer, using multi-configured 1D CNN to perform ensemble methods at each layer, and using the grid search technique to determine the optimal combination of weights for each layer.

The benefits of the model that we have developed are as follows. This model can carry out the classification process from one layer to another by expanding its learning knowledge. During the classification phase, the learning knowledge is improved from its internal resources. Each layer determines the ideal weighting combination to produce the best results. Finally, the predictions from multiple CNN models are combined to reduce the variance of predictions and reduce generalization error. The limitations of our proposed model are as follows. The computational time is high for a high dimensional dataset. Lots of training data is required. Furthermore, it cannot encode the position or orientation of a DNA sequence and, finally, it needs more memory space.

## 5. Conclusions and Future Work

DNA 4mC is a crucial epigenetic modification that causes various diseases and is also a restriction–modification system. Therefore, accurately identifying 4mC sites is an essential step towards understanding the exact biological functions. This work presented an influential computational model named GS-MLDS for predicting 4mC and non-4mC sites. We have proposed a reusable knowledge model that aids in the transmission of a previous layer’s knowledge to the next layer. This study finds the optimal accuracy of each layer by applying a grid-based search technique to find the best weight set combinations in each layer. This study has experimented with eight different types of datasets. With an 80:20 split between the training and testing sets, our proposed model attained accuracy levels of 0.978, 0.954, 0.944, 0.961, 0.950, 0.973, 0.978, 0.954, 0.948, 0.952, 0.953, and 0.980. The correct classification probability of 4mC and non-4mC of the proposed system has been pointed out by the AUC curve. That is, the proposed model has 96.97%, 94.35%, 92.30%, 95.51%, 98.07%, 96.49%, 95.49%, and 96.70% probability of distinguishing 4mC from non-4mC for each dataset, respectively. The suggested methodology has shown improved performance compared to other machine learning models. Our proposed approach outperformed the state-of-the-art predictors for each dataset in terms of detecting 4mC sites in both balanced and unbalanced class labels. The study presented in the paper may be useful for more pervasive bioinformatics applications. The goal of this research’s future work is to produce more semantic characteristics and create a model that has already been trained on a big dataset to address a similar issue.

## Figures and Tables

**Figure 1 genes-14-00582-f001a:**
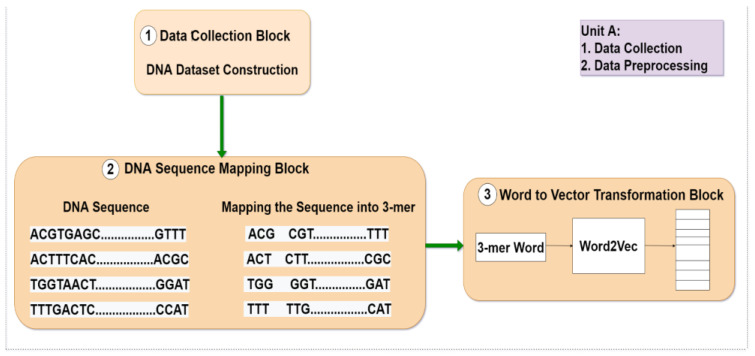
(**a**) (Unit A) Dataset contraction, feature representation, and distributed feature representation block.

**Figure 1 genes-14-00582-f001b:**
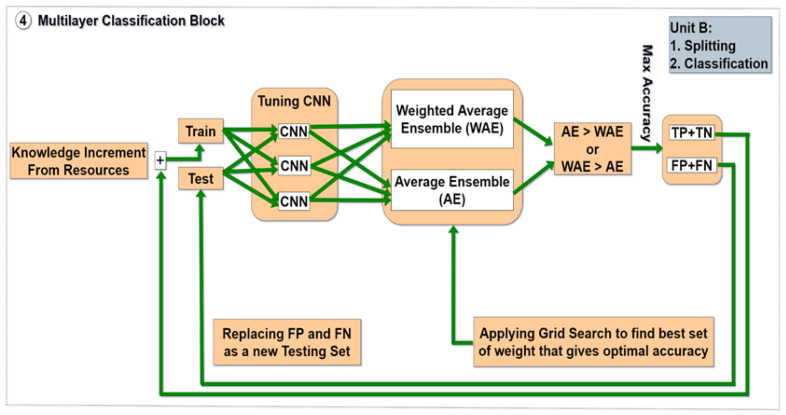
(**b**) (Unit B (Classification Block)) Grid search-based multilayer dynamic system (GS-MLDS) to predict 4mC sites.

**Figure 2 genes-14-00582-f002:**
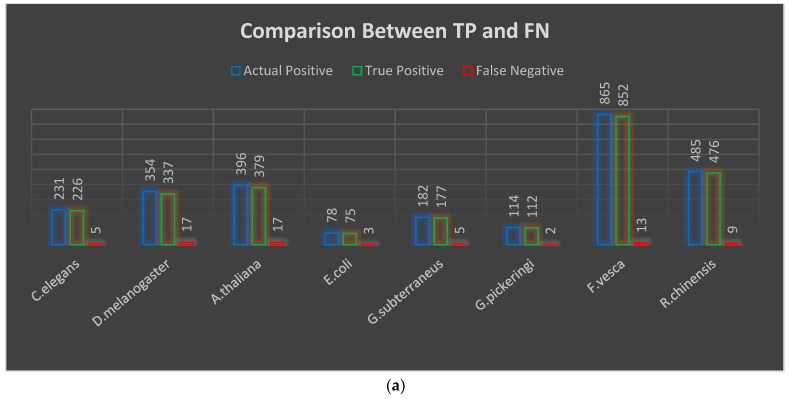
(**a**) Comparison between TP and FN, (**b**) Comparison between TN and FP.

**Figure 3 genes-14-00582-f003:**
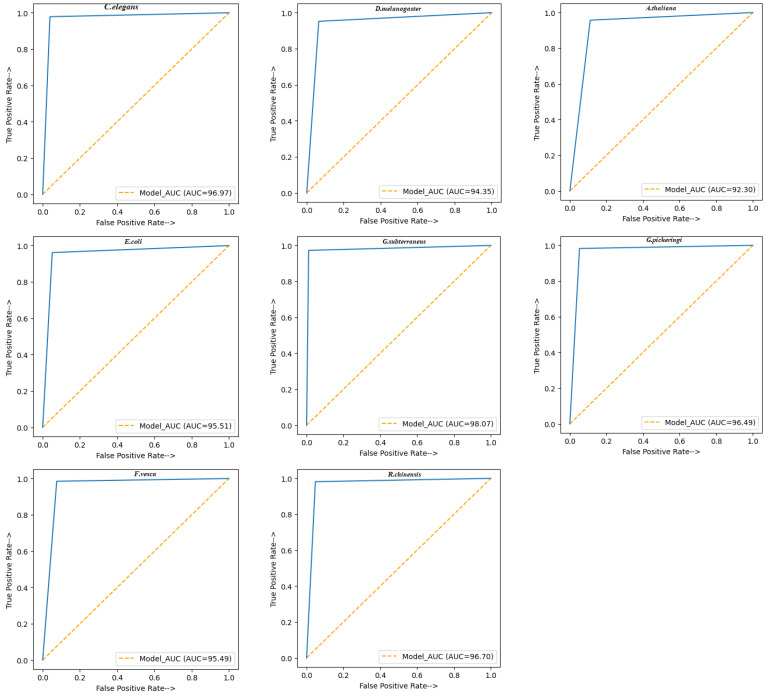
ROC curve and AUC of the proposed model.

**Table 1 genes-14-00582-t001:** Brief description of related studies adopted from the Refs. [3,5,6,7,8,9,10,11,12,13,15,16,17,18,19].

Authors	Contributions	Datasets	Performances
S. Zhang et al. (2022) [10]	1. Applied three types of feature extraction methods to extract sequence information. 2. Used a gradient boosting decision tree (GBDT) to select the important features and remove redundant information. 3. Used six benchmark datasets to test the performance of the model.	*C. elegans* (3108 samples)	Acc: 0.851, Mcc: 0.703, Sn: 0.872, Sp: 0.83
		*D. melanogaster* (3538 samples)	Acc: 0.859, Mcc: 0.717, Sn: 0.868, Sp: 0.849
		*A. thaliana* (3956 samples)	Acc: 0.801, Mcc: 0.602, Sn: 0.793, Sp: 0.81
		*E. coli* (776 samples)	Acc: 0.881, Mcc: 0.763, Sn: 0.886, Sp: 0.877
		*G. subterraneus* (1812 samples)	Acc: 0.859, Mcc: 0.719, Sn: 0.862, Sp: 0.856
		*G. pickeringi* (1138 samples)	Acc: 0.901, Mcc: 0.802, Sn: 0.898, Sp: 0.905
Yu. Lezheng et al. (2022) [11]	1. Focused on three types of deep learning architectures: convolutional neural networks (CNNs), recurrent neural networks (RNNs), and convolutional recurrent neural networks (CNN-RNNs). 2. Analyzed several important factors, such as model architecture and its hyperparameters (the number of filters, kernel, pooling, and BiLSTM sizes, etc.), encoding methods, and attention mechanisms. 3. Introduced UMAP and Deep SHAP to better analyze and understand deep learning models.	*C. elegans* (3108 samples)	Acc: 0.839, Mcc: 0.678, Recall: 0.84
		*D. melanogaster* (3538 samples)	Acc: 0.894, Mcc: 0.789, Recall: 0.903
		*A. thaliana* (3956 samples)	Acc: 0.874, Mcc: 0.749, Recall: 0.896
L. Wang et al. (2022) [12]	1. Fuzzy support vectors were used to design the machine learning model. 2. Trinucleotide positional preference was used to convert the DNA sequence into a numerical vector that fully extracts the information from the benchmark dataset. 3. Analyzed model architecture and hyperparameters (for example, the number of filters, kernel, pooling, and BiLSTM sizes), encoding methods, and attention mechanisms. 4. Applied a grid search algorithm to find the best parameters.	*C. elegans* (3108 samples)	Acc: 0.875, Mcc: 0.750, Sn: 0.873, Sp: 0.876
		*D. melanogaster* (3538 samples)	Acc: 0.871, Mcc: 0.743, Sn: 0.874, Sp: 0.868
		*A. thaliana* (3956 samples)	Acc: 0.828, Mcc: 0.756, Sn: 0.824, Sp: 0.834
		*E. coli* (776 samples)	Acc: 0.962, Mcc: 0.923, Sn: 0.962, Sp: 0.961
		*G. subterraneus* (1812 samples)	Acc: 0.901, Mcc: 0.804, Sn: 0.909, Sp: 0.893
		*G. pickeringi* (1138 samples)	Acc: 0.911, Mcc: 0.824, Sn: 0.914, Sp: 0.910
J. Jin et al. (2022) [13]	1. The adaptive embedding approach that has been proposed may automatically change the original input feature to better represent the prediction task. 2. An effective network of bidirectional gated recurrent units was applied to efficiently extract the complete and meaningful representation of the full DNA sequence from both near and far information.	*G. subterraneus* (1812 samples)	Acc: 0.825, Mcc: 0.651, Sn: 0.8, Sp: 0.85
J. Khanal et al. (2021) [3]	1. Applied a word embedding method to capture the high-level input features. 2. Double-layer 1D CNN was applied to process the captured features.	*F. vesca* (25,922 samples),	Acc: 0.869, Auc: 0.940, Sn: 0.897, Sp: 0.860
		*R. chinensis* (14,502 samples)	Acc: 0.854, Auc: 0.937, Sn: 0.871, Sp: 0.885
A. Wahab et al. (2021) [5]	1. Used word2vec embedding to transfer the sequences into the form of vectors. 2. Fed the vectors of word embedding into CNN by applying grid search-based algorithm.	*C. elegans*(12,726 samples).	Acc: 0.935, Auc: 0.973, Sn: 0.956, Sp: 0.899
G. Fang et al. (2021) [6]	1. Used word2vec encoding and one-hot encoding to conduct comparative tests on the same dataset. 2. Designed a deep learning framework to combine a three-CNN module to extract the hidden high-level features and more biological features.	*C. elegans*(17,808 samples).	Acc: 0.932, Auc: 0.971, Sn: 0.950, Sp: 0.916
H. Zulfiqar et al. (2021) [7]	1. Applied a word embedding method to capture the high-level input features. 2. Double-layer 1D CNN was applied to process the captured features on the basis of 10-fold cross-validation.	*E. coli* (776 samples)	Acc: 0.861, Mcc: 0.670, Sn: 0.876, Sp: 0.773
D.Y. Lim et al. (2021) [15]	1. Applied LSTM to solve the vanishing gradient problem by adding extra interactions. 2. Hyper-parameter tuning was applied to optimize filter sizes, number of filters, number of convolution layers, pool sizes, stride length, and dropout values.	*F.vesca* (12,942 samples)	Acc: 0.859, Mcc: 0.706, Sn: 0.835, Sp: 0.882
		*R. chinensis* (6232 samples)	Acc: 0.865, Mcc: 0.714, Sn: 0.854, Sp: 0.875
M. Tahir et al. (2021) [9]	1. Applied “word2vec” to automatically extract features from DNA sequences and feed the vectors into CNN. 2. Compared to the state-of-the-art methods for all six benchmark datasets and independent datasets.	*C. elegans* (3108 samples)	Acc: 0.87, Auc: 0.937, Sn: 0.887, Sp: 0.857
		*D. melanogaster* (3538 samples)	Acc: 0.882, Auc: 0.938, Sn: 0.894, Sp: 0.870
		*A. thaliana* (3956 samples)	Acc: 0.840, Auc: 0.902, Sn: 0.894, Sp: 0.829
		*E. coli* (776 samples)	Acc: 0.868, Auc: 0.946, Sn: 0.873, Sp: 0.866
		*G. subterraneus* (1812 samples)	Acc: 0.886, Auc: 0.949, Sn: 0.888, Sp: 0.884
		*G. pickeringi* (1138 samples)	Acc: 0.925, Auc: 0.968, Sn: 0.912, Sp: 0.938
M.M. Hasan et al. (2020) [8]	1. Employed six encoding schemes to cover various aspects of DNA sequences. 2. Optimized the successive features via the WR feature selection method.	*G. subterraneus* (1812 samples)	Acc: 0.816, Auc: 0.920, Sn: 0.807, Sp: 0.825
A. Wahab el al. (2020) [16]	1. Applied six encoding techniques. 2. Applied a grid search algorithm to obtain the optimal model.	*F. vesca* (12,942 samples)	Acc: 0.815, Auc: 0.89, Sn: 0.878, Sp: 0.753
		*R. chinensis* (6232 samples)	Acc: 0.783, Auc: 0.87, Sn: 0.801, Sp: 0.765
		*Cross-species* (17,722 Samples)	Acc: 0.780, Auc: 0.85, Sn: 0.849, Sp: 0.706
Z. Zhao et al. (2020) [17]	1. Applied embedded feature selection scheme to rank with the feature relevance scores. 2. Extracted the sequence properties of contiguous nucleotides as features to characterize the sequences.	*C. elegans* (3108 samples),	Acc: 0.826, Mcc: 0.653, Sn: 0.849, Sp: 0.804
		*D. melanogaster* (3538 samples)	Acc: 0.843, Mcc: 0.686, Sn: 0.854, Sp: 0.832
		*A. thaliana* (3956 samples)	Acc: 0.794, Mcc: 0.589, Sn: 0.783, Sp: 0.805
		*E. coli* (776 samples)	Acc: 0.843, Mcc: 0.686, Sn: 0.861, Sp: 0.825
		*G. subterraneus* (1812 samples)	Acc: 0.847, Mcc: 0.694, Sn: 0.836, Sp: 0.857
		*G. pickeringi* (1138 samples).	Acc: 0.877, Mcc: 0.754, Sn: 0.863, Sp: 0.891
J. Khanal et al. (2019) [18]	1. Selected the best performing hyper-parameters using the grid search method. 2. Extracted the features of the 4mC sites from DNA sequence automatically using the CNN model.	*C. elegans* (3108 samples)	Acc: 0.842, Mcc: 0.694, Sn: 0.894, Sp: 0.825
		*D. melanogaster* (3538 samples)	Acc: 0.853, Mcc: 0.686, Sn: 0.864, Sp: 0.853
		*A. thaliana* (3956 samples),	Acc: 0.797, Mcc: 0.621, Sn: 0.803, Sp: 0.792
		*E. coli* (776 samples)	Acc: 0.859,Mcc: 0.687, Sn: 0.881, Sp: 0.788
		*G. subterraneus* (1812 samples)	Acc: 0.860, Mcc: 0.703, Sn: 0.851, Sp: 0.843
		*G. pickeringi* (1138 samples)	Acc: 0.871, Mcc: 0.750, Sn: 0.857, Sp: 0.893
B. Manavalan et al. (2019) [19]	1. Applied a feature representation learning scheme and generated 56 probabilistic features based on 4 different ML algorithms and 7 feature encodings covering diverse sequence information, including compositional, physicochemical, and NT position-specific information. 2. Compared the performance of the proposed predictor with those of three state-of-the art predictors.	*C. elegans* (3108 samples)	Acc: 0.826, Auc: 0.892, Sn: 0.840, Sp: 0.812
		*D. melanogaster* (3538 samples)	Acc: 0.842, Auc: 0.904, Sn: 0.831, Sp: 0.854
		*A. thaliana* (3956 samples)	Acc: 0.792, Auc: 0.861, Sn: 0.761, Sp: 0.822
		*E. coli* (776 samples)	Acc: 0.848, Auc: 0.911, Sn: 0.869, Sp: 0.827
		*G. subterraneus* (1812 samples)	Acc: 0.855, Auc: 0.856, Sn: 0.854, Sp: 0.904
		*G. pickeringi* (1138 samples)	Acc: 0.891, Auc: 0.951, Sn: 0.884, Sp: 0.898

**Table 2 genes-14-00582-t002:** The number of positive and negative samples collected from each of the eight species.

Dataset Name	$S_{i}^{+}$	$S_{i}^{-}$	$S_{i}=S_{i}^{+}\cup S_{i}^{-}$
*Caenorhabditis elegans (C. elegans)*	1154	1154	2308
*Drosophila melanogaster (D. melanogaster)*	1769	1769	3538
*Arabidopsis thaliana (A. thaliana)*	1978	1978	3956
*Escherichia coli (E. coli)*	388	388	776
*Geoalkalibacter subterraneus (G. subterraneus)*	906	906	1812
*Geobacter pickeringii (G. pickeringi)*	569	569	1165
*Fragaria vesca (F. vesca)*	4321	4321	8642
*Rosa chinensis (R. chinensis)*	2421	2421	4842

**Table 3 genes-14-00582-t003:** Training parameters used in word2vec.

Parameters	word2vec Model
Training Method	CBOW
Corpus	*C. elegans*, *D. melanogaster*, *A. thaliana*, *E. coli*, *G. subterraneus*, *G. pickeringi*, *F. vesca*, and *R. chinensis.*
Context word	3-mer
Vector size	100
Window size	5
Minimum count	5
Negative sampling	5
Epochs	25

**Table 4 genes-14-00582-t004:** Hyper-parameters tuning demonstration.

Parameters	Range
The number of convolutional layers	(1, 2, 3, 4, 5)
The number of filters in each convolutional layer	(16, 32, 42, 64)
The size of filters of the convolutional layers	(3, 5, 7, 9)
Dropout rate	(0.2, 0.4, 0.5, 0.6)

**Table 5 genes-14-00582-t005:** The detailed architecture of the proposed CNN.

Layer (Model: 1)	Layer (Model: 2)	Layer (Model: 3)
Input; Shape (39, 100)	Input; Shape (39, 100)	Input; Shape (39, 100)
Conv 1D (16, 3, 1)	Conv 1D (32, 5, 1)	Conv 1D (42, 7, 1)
Conv 1D (32, 5, 1)	Conv 1D (42, 7, 1)	Conv 1D (64, 9, 1)
Dropout (0.5)	Dropout (0.3)	Dropout (0.6)
Dense (1)	Dense (1)	Dense (1)
Sigmoid	Sigmoid	Sigmoid

**Table 6 genes-14-00582-t006:** Performance evaluation metrics result from the proposed approach (GS-MLDS) based on a different datasets.

Dataset Name	Parameters (%)						
	ACC	AUC	Precision	Recall /TRP	FPR	TNR	FNR
*C. elegans*	96.96	96.97	96.17	97.83	3.89	96.10	2.16
*D. melanogaster*	94.35	94.35	93.61	95.19	6.49	93.50	4.80
*A. thaliana*	92.29	92.30	89.59	95.70	11.11	88.88	4.29
*E. coli*	95.51	95.51	94.93	96.15	5.12	94.87	3.84
*G. subterraneus*	98.07	98.07	98.88	97.25	1.10	98.89	2.74
*M. pickeringi*	96.49	96.49	94.91	98.24	5.26	94.73	1.75
*F. vesca*	95.48	95.49	92.91	98.49	7.52	92.47	1.50
*R. chinensis*	96.69	96.70	95.39	98.14	4.75	95.24	1.85

**Table 7 genes-14-00582-t007:** Layer to layer classification performance of GS-MLDS.

Dataset Name	Layer No.	$\mathrm{Train}\;\mathrm{Data},\;x_{i}$ = $x_{i-1}+{\left(TP,TN\right)}_{i-1}$	$\mathrm{Test}\;\mathrm{Data},\;y_{i}$ $={\left(FP,FN\right)}_{i-1}$	Method					Total Accuracy, $\overset{N}{\underset{i=1}{\sum }}\frac{{\left(TP,TN\right)}_{i}}{initial,\;y}$
				AE (Average Ensemble)		WAE (Weighted Average Ensemble)			
				TP|TN	FP|FN	TP|TN	FP|FN	Weights	
*C. elegans* (17,808 Instances)	1	1846	462	124	338	196	266	0.3, 0.1, 0.2	42.42
	2	2042	266	78	260	118	148	0.0, 0.1, 0.3	67.96
	3	2160	148	23	237	88	60	0.0, 0.1, 0.2	87.01
	4	2248	60	12	225	46	14	0.1, 0.5, 0.2	**96.96**
*D. melanogaster* (3538 Instances)	1	2830	708	362	346	384	324	0.2, 0.1, 0.3	54.23
	2	3214	384	194	152	197	127	0.2, 0.1, 0.0	82.06
	3	3411	197	66	86	87	40	0.0, 0.3, 0.2	**94.35**
*A. thaliana* (3956 Instances)	1	3164	792	321	471	337	455	0.0, 0.4, 0.2	42.55
	2	3501	425	103	368	227	228	0.1, 0.3, 0.7	71.21
	3	3728	198	87	281	167	61	0.3, 0.5, 0.2	**92.29**
*E. coli* (776 Instances)	1	620	156	87	69	110	46	0.3, 0.5, 0.2	70.51
	2	730	46	37	32	39	7	0.2, 0.1, 0.4	**95.51**
*G. subterraneus* (1812 Instances)	1	1449	363	123	240	200	163	0.1, 0.0, 0.3	55.00
	2	1649	64	55	185	99	64	0.1, 0.2, 0.5	82.36
	3	1748	7	36	149	57	7	0.1, 0.0, 0.2	**98.07**
*G. pickeringi* (1165 Instances)	1	910	228	106	122	108	120	0.4, 0.1, 0.2	47.36
	2	1018	120	67	55	77	43	0.0, 0.2, 0.1	81.14
	3	1095	43	22	33	35	8	0.4, 0.2, 0.6	**96.49**
*F. vesca* (25,922 Instances)	1	6913	1729	857	872	1184	545	0.2, 0.1, 0.3	68.47
	2	8097	545	127	745	343	202	0.1, 0.4, 0.6	88.31
	3	8440	202	53	692	124	78	0.1, 0.0, 0.2	**95.48**
*R. chinensis* (14,502 Instances)	1	3837	969	353	616	689	280	0.1, 0.4, 0.2	71.10
	2	4526	280	157	459	248	32	0.5, 0.3, 0.2	**96.69**

**Table 8 genes-14-00582-t008:** Comparison of GS-MLDS with another author’s model utilizing their datasets.

Authors	Splitting Ratio	Dataset Name	Authors Model Accuracy	GS-MLDS Accuracy
S. Zhang et al. (2022) [10]	10-fold	*C. elegans*	0.851	0.978
		*D. melanogaster*	0.859	0.954
		*A. thaliana*	0.801	0.944
		*E. coli*	0.870	0.961
		*G. subterraneus*	0.859	0.950
		*G. pickeringi*	0.901	0.973
Yu. Lezheng et al. (2022) [11]	10-fold	*C. elegans*	0.894	0.978
		*D. melanogaster*	0.874	0.954
		*A. thaliana*	0.839	0.944
L. Wang et al. (2022) [12]	10-fold	*C. elegans*	0.875	0.978
		*D. melanogaster*	0.871	0.954
		*A. thaliana*	0.828	0.944
		*E. coli*	0.962	0.961
		*G. subterraneus*	0.901	0.950
		*G. pickeringi*	0.911	0.973
S. Zhang et al. (2022) [20]	10-fold	*C. elegans*	0.851	0.978
		*D. melanogaster*	0.859	0.954
		*A. thaliana*	0.801	0.944
		*E. coli*	0.87	0.961
		*G. subterraneus*	0.859	0.950
		*G. pickeringi*	0.901	0.973
J. Jin et al. (2022) [13]	10-fold	*G. subterraneus*	0.825	0.950
J. Khanal et al. (2021) [3]	5-fold	*F. vesca*	0.869	0.948
		*R. chinensis*	0.854	0.952
G. Fang et al. (2021) [6]	3-fold	*C. elegans*	0.932	0.961
A. Wahab et al. (2021) [5]	3-fold	*C. elegans*	0.935	0.961
M. Tahir et al. (2021) [9]	10-fold	*C. elegans*	0.872	0.978
		*D. melanogaster*	0.882	0.954
		*A. thaliana*	0.840	0.944
		*E. coli*	0.868	0.961
		*G. subterraneus*	0.886	0.950
		*G. pickeringi*	0.925	0.973
H. Zulfiqar et al. (2021) [14]	10-fold	*E. coli*	0.861	0.961
D.Y. Lim et al. (2021) [15]	10-fold	*F. vesca*	0.859	0.953
		*R. chinensis*	0.865	0.953
A. Wahab el al. (2020) [16]	10-fold	*F. vesca*	0.815	0.953
		*R. chinensis*	0.783	0.953
Z. Zhao et al. (2020) [17]	10-fold	*C. elegans*	0.826	0.978
		*D. melanogaster*	0.842	0.954
		*A. thaliana*	0.794	0.944
		*E. coli*	0.843	0.961
		*G. subterraneus*	0.847	0.950
		*G. pickeringi*	0.877	0.973
M.M. Hasan et al. (2020) [8]	80:20	*G. subterraneus*	0.816	0.980
J. Khanal et al. (2019) [18]	10-fold	*C. elegans*	0.842	0.978
		*D. melanogaster*	0.853	0.954
		*A. thaliana*	0.797	0.944
		*E. coli*	0.859	0.961
		*G. subterraneus*	0.860	0.950
		*G. pickeringi*	0.871	0.973
B. Manavalan et al. (2019) [19]	10-fold	*C. elegans*	0.826	0.978
		*D. melanogaster*	0.842	0.954
		*A. thaliana*	0.792	0.944
		*E. coli*	0.848	0.961
		*G. subterraneus*	0.855	0.950
		*G. pickeringi*	0.891	0.973

## Data Availability

All datasets are available at the link below: https://github.com/rajib1346/GS-MLDS.git; and are available from the corresponding author upon request (2 November 2022).

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
