# Peer review of "A Grid Search-Based Multilayer Dynamic Ensemble System to Identify DNA N4—Methylcytosine Using Deep Learning Approach"

_genes, 2023, doi:10.3390/genes14030582_

Round 1

Reviewer 1 Report

- I think that the literature review part of the study is given in detail. However, I think an addition should be made that can increase the interest of the readers. The presence of a literature summary table of similar studies in this section will make the study easier to understand by the readers. (table column headings = method, data set, results obtained, success rate, etc. may consist of information).

- Why was the trinucleotide composition (k=3) chosen? Please explain.

- It may be more accurate to present the proposed algorithm as an addendum.

- The difference of the study from the previous studies has been revealed, thus emphasizing the originality of the research. However, I believe that it is not necessary to give the journal and publisher information. Already at the end of the sentence, a reference is made to the relevant work. (Related sentence: Based on experimental results, we mentioned in a journal named "Informatics in Medicine Unlocked," published by Elsevier in 2021)

- When I scan the title of the study in google scholar, I see a link to the study. Have you previously published this study in another journal? Why did such a situation occur?

Reviewer 2 Report

The title “A Grid Search-Based Multilayer Dynamic Ensemble System to Identify DNA N4– Methylcytosine Using Deep Learning Approachis quite interesting but I have some concerns on this work which are given below.

1: In abstract part line 18, it is critical to precisely locate the 4mC site and detect its chromosomal distribution. The word critical is not suitable here please revise this sentence.

2: In line 51, Please correct the references. In line 67-76 please use either bullets or numbers. In line 127 remove italics. In line 133 and 142, remove spaces and in line 146 make space C.elegans(17808 samples).

3: From line 142-145, please revise this paragraphs carefully. G. Fang et al. (2021) [6] proposed a Word2vec based deep learning network for DNA N4-methylcytosine sites identification. Each feature in the vector derived from K-mers (K= K = 3, 4, 5, 6) was presented using the word2vec embedding approach. The feature matrix sequence was then feed into 3-CNN to categorize the 4mCs and non-4mCs. Dataset name: C.elegans(17808 samples).

4: Figure 1 is not explained properly. Please correct this. Figure 1. (a). (Unit: A) Proposed grid search-based multilayer dynamic system (GS-MLDS) to predict 4mC sites. (b). (Unit: B) Proposed grid search-based multilayer dynamic system (GS-MLDS) to predict 4mC sites.

5: From line 454-466, Please write the advantages and limitations in the form of paragraph.

6: Add some more data and citations in conclusion part.

7: Please correct the references specially reference no. 14. Reference no. 7 and 15 are also same. I’ll recommend you to add these citations regarding your topic in this reference list (Deep-4mCGP: A Deep Learning Approach to Predict 4mC Sites in Geobacter pickeringii by Using Correlation-Based Feature Selection Technique https://doi.org/10.3390/ijms23031251, Advances in mapping the epigenetic modifications of 5-methylcytosine (5mC), N6-methyladenine (6mA), and N4-methylcytosine (4mC) https://doi.org/10.1002/bit.27911,  Computational identification of N4-methylcytosine sites in the mouse genome with machine-learning method https://doi.org/10.3934/mbe.2021167 ).

Round 2

Reviewer 1 Report

The researchers made the necessary corrections in the study. The Manuscript can be published as such.